# Dietary Fishmeal Can Be Partially Replaced with Non-Grain Compound Proteins through Evaluating the Growth, Biochemical Indexes, and Muscle Quality in Marine Teleost *Trachinotus ovatus*

**DOI:** 10.3390/ani13101704

**Published:** 2023-05-21

**Authors:** Zeliang Su, Yongcai Ma, Fang Chen, Wenqiang An, Guanrong Zhang, Chao Xu, Dizhi Xie, Shuqi Wang, Yuanyou Li

**Affiliations:** 1University Joint Laboratory of Guangdong Province, Hong Kong and Macao Region on MBCE, College of Marine Sciences, South China Agricultural University, Guangzhou 510642, China; zeliangsu@stu.scau.edu.cn (Z.S.); mayc1992@163.com (Y.M.); chenfang@scau.edu.cn (F.C.); anwenqiang0314@163.com (W.A.); zhanggr@stu.scau.edu.cn (G.Z.); xuc1213@scau.edu.cn (C.X.); xiedizhi@scau.edu.cn (D.X.); 2Guangdong Provincial Key Laboratory of Marine Biotechnology, Shantou University, Shantou 515063, China; sqw@stu.edu.cn

**Keywords:** fishmeal replacement, growth performance, non-grain protein, protein sources, serum biochemical indexes, textural properties, *Trachinotus ovatus*

## Abstract

**Simple Summary:**

The application of more inedible ingredients in aquafeeds will contribute to sustainable development and alleviate food security. The rising prices and declining yields of traditional aquafeed protein sources such as fishmeal and soybean meal are hindering the development of aquaculture. Therefore, in this study, bovine bone meal, dephenolized cottonseed protein, and blood cell meal were utilized to replace dietary fishmeal in the form of compound protein. Culture trials were conducted on the economically farmed fish, juvenile golden pompano, and the feasibility and suitable ratio of non-grain proteins for application in aquafeeds were evaluated by indicators such as growth performance and muscle quality. The results of this study provide data support for the development of new aquafeeds with less fishmeal and high efficiency, which can help alleviate the shortage of aquafeed feedstuff, and provide theoretical and data references for the study of securing a sustainable food supply.

**Abstract:**

In the context of human food shortages, the incorporation of non-grain feedstuff in fish feed deserves more research attention. Here, the feasibility and appropriate ratio of non-grain compound protein (NGCP, containing bovine bone meal, dephenolized cottonseed protein, and blood cell meal) for dietary fishmeal (FM) replacement were explored in golden pompano (*Trachinotus ovatus*). Four isonitrogenous (45%) and isolipidic (12%) diets (Control, 25NGP, 50NGP, and 75NGP) were prepared. Control contained 24% FM, whereas the FM content of 25NGP, 50NGP, and 75NGP was 18%, 12%, and 6%, respectively, representing a 25%, 50%, and 75% replacement of FM in Control by NGCP. Juvenile golden pompano (initial weight: 9.71 ± 0.04 g) were fed the four diets for 65 days in sea cages. There was no significant difference between the 25NGP and Control groups in terms of weight gain, weight gain rate, and specific growth rate; contents of crude protein, crude lipid, moisture, and ash in muscle and whole fish; muscle textural properties including hardness, chewiness, gumminess, tenderness, springiness, and cohesiveness; and serum biochemical indexes including total protein, albumin, blood urea nitrogen, HDL cholesterol, total cholesterol, and triglycerides. However, the golden pompano in 50NGP and 75NGP experienced nutritional stress, and thus some indicators were negatively affected. In addition, compared to the Control group, the expression levels of genes related to protein metabolism (*mtor*, *s6k1*, and *4e-bp1*) and lipid metabolism (*pparγ*, *fas*, *srebp1*, and *acc1*) of the 25NGP group showed no significant difference, but the *4e-bp1* and *pparγ* of the 75NGP group were significantly upregulated and downregulated, respectively (*p* < 0.05), which may explain the decline in fish growth performance and muscle quality after 75% FM was replaced by NGCP. The results suggest that at least 25% FM of Control can be replaced by NGCP, achieving a dietary FM content of as low as 18%; however, the replacement of more than 50% of the dietary FM negatively affects the growth and muscle quality of golden pompano.

## 1. Introduction

The global population is estimated to grow by 50% by 2050, doubling the demand for animal protein; the existing food productivity will not be able to meet such an enormous demand [1]. Aquaculture is an essential component of food production, and aquaculture species represent a source of nutrition and healthy proteins for humans [2]. Aquatic animals tend to have higher feed efficiency than livestock and poultry, which facilitates a higher feed conversion ratio and contributes to a sustainable supply of protein for humans in the future [3]. Due to its rich nutritional value and balanced amino acid composition, fishmeal (FM) is often added to aquafeeds, especially feeds for carnivorous marine teleosts [4]. However, in recent years, with the depletion of wild fishery resources worldwide, the production of capture fisheries has been decreasing every year, whereas the production of FM has decreased, and its price has increased [5]. Currently, soybean meal and corn gluten meal are the traditional feed protein sources being widely used in feed production [6]. However, in the future, grain products such as soybean meal and corn, which are sources of human edible protein (HEP), may be more required for human consumption [7]. HEP is defined as protein that has a sufficiently high nutritional value and can be consumed by humans; humans are in competition with animals for the use of HEP sources [7]. It has been reported that approximately 60% of globally produced protein is used for animal and fish feed, and a large range of underutilized non-food proteins are worthy of full consideration as alternative proteins [8]. In addition, the increasing prices and decreasing availability of FM and soybean necessitate the search for additional protein sources with high nutritional value and low cost for aquafeeds [9,10]. The incorporation of more non-grain proteins in aquatic feeds would help restructure the current feed composition and alleviate the potential food crisis.

In fact, many by-products of animal and plant production have a high nutritional value, but are not fully utilized in aquatic feeds. For example, cottonseed meal, a by-product of oilseed crops and fibers (55–68% protein content) [11], is the common protein source for poultry feed [12]. In recent years, it has received attention as an alternative protein to FM in aquafeeds, such as rainbow trout (*Oncorhynchus mykiss*) [13], large yellow croaker (*Larimichthys crocea*) [14], sturgeon (*Acipenser schrenckii*) [15], etc., but due to the toxic effects of free cottonseed on fish, the high percentage of addition may affect the growth and health of fish. According to statistics, in 2020, the global meat production was approximately 252.6 billion kg, including 57.7 billion kg of beef, 99.1 billion kg of chicken, and 95.8 billion kg of pork [16], while the meat industry produces several by-products during slaughter and processing, such as blood, bones, skin, organs, internal organs, horns, hooves, feet, and skulls [17]. Among them, blood meal—a product of fresh blood obtained from slaughtered food animals processed using high-temperature cooking, sterilization, and drying—has a protein content of 90–95% [18]. Currently, blood meal has been reported In FM replacement studies for totoaba (*Totoaba macdonaldi*) [19], rainbow trout [20], and Atlantic salmon (*Salmo salar* L.) [21], and its low digestibility is the main limiting factor for its application. In addition, bovine bone meal, a type of meat and bone meal—obtained from livestock by-products such as unconsumable bones using heating, drying, and crushing—has the advantages of high protein content and low price [22]. However, meat and bone meal has also been reported to have low apparent digestibility in mandarin fish (*Siniperca chuatsi*) [23]. In general, the above reports demonstrate that the high substitution of these proteins for fishmeal in aquafeeds reduces the growth and muscle quality of fish and even endangers their health. Therefore, how to rationally use these non-grain protein sources to replace FM in aquatic formula feed has become a key problem in FM substitution applications.

The golden pompano (*Trachinotus ovatus*), a carnivorous marine teleost, is mainly distributed in the Indian Ocean, China Sea, off the coasts of Indonesia, Australia, and Japan, tropical and temperate Atlantic waters of America, and off the west coast of Africa [24]. Because of its high nutritional value, stress resistance, and survival rate, golden pompano is widely farmed along the southern coast of East Asia [25], and its annual production in China is more than 150 million kg [26]. The aquaculture of this fish is highly dependent on FM, and the FM content in its commercial feed is typically more than 30%, which greatly limits the sustainability and economic benefits of its aquaculture [24]. Therefore, many studies have focused on FM substitution in golden pompano feed. Wu et al. reported that the use of soybean protein concentrate, together with supplementation with selenium yeast to the feed, can reduce the percentage of FM to 24% [27]. Shen et al. discovered that dietary FM could be reduced to 13.60% with cottonseed protein concentrate without affecting the weight gain rate [28]. However, Fu et al. found that the replacement of more than 20% of FM in the diet by low-gossypol cottonseed meal resulted in impaired intestinal barrier function in golden pompano [29]. Similarly, Qin et al. found that the use of cottonseed meal could only reduce the feed FM content to 20% without affecting the growth of golden pompano [30] and that the fish muscle nutritive deposition was negatively affected when the feed FM was below 15% [31]. The several different results mentioned above may correlate with the level of detoxification of cottonseed protein. Notably, a previous study in our laboratory found that the high percentage of FM in the diet of golden pompano could be substituted with a terrestrial compound protein, reducing the FM content to 6% without negatively affecting growth performance [32]. This finding suggests that compound protein replacing FM in aquatic feed has a better culture effect than single protein replacing FM, which prompted us to investigate whether non-grain compound protein (NGCP) could be used in the feed of golden pompano, which may help solve the current problem of protein shortage in aquafeed.

In the present study, a source of NGCP was formulated to replace dietary FM in golden pompano feed in different proportions. After a 65-day rearing trial, the feasibility of application and suitable ratio of this NGCP were determined by evaluating the growth index, proximate composition, serum biochemical indexes, muscle quality, and protein and lipid metabolism gene mRNA expression levels of fish in each group. In this study, three common non-food proteins (bovine bone meal, dephenolized cottonseed protein, and blood cell meal) were innovatively applied to marine carnivorous fish feeds in the form of compound proteins, and the possible mechanisms for the differences in growth and muscle quality of cultured animals were analyzed from the perspectives of muscle protein metabolism and lipid metabolism.

## 2. Materials and Methods

### 2.1. Experimental Diets

Here, a NGCP containing dephenolized cottonseed protein, bovine bone meal, and blood cell meal (in the ratio 5:3:2) was used as an ingredient in three of four isonitrogenous (45%) and isolipidic (12%) diets (Control, 25NGP, 50NGP, and 75NGP), replacing 0%, 25%, 50%, and 75% of FM in each diet, respectively. Control contained 24% FM, whereas the FM in 25NGP, 50NGP, and 75NGP was reduced to 18%, 12%, and 6%, respectively, owing to its replacement with NGCP. The basic protein sources in each diet were soybean protein concentrate, poultry meal, corn gluten meal, and fermented soybean meal. The lipid source in each diet was an oil blend designed for golden pompano feed previously developed in our laboratory, mainly formulated with fish oil, soybean oil, rapeseed oil, perilla oil, phospholipids, and small quantities of emulsifiers and antioxidants [33]. The specific ratios and basic nutrition composition of the four diets have been shown in Table 1. The preparation process of diets and the equipment used in this study were according to Ma et al. [32]. After pelleting, all feeds were placed in a room at 17 °C for about 3 days until dried, then sealed and stored in a freezer at 20 °C.

The dietary amino acid composition and fatty acid profile are shown in Table 2 and Table 3, respectively.

### 2.2. Animal and Breeding Management

The juvenile golden pompano (approximately 2 g) for this experiment were purchased from a local fish hatchery (Shantou, China) and subsequently temporarily housed in an offshore net (2 m × 2 m × 1.5 m. L/W/H) at the Nan’ao Marine Biological Station (NAMBS) of Shantou University until trials. During the acclimatization, the juvenile fish were fed with commercial feed (Guangdong Yuehai Feed Co., Ltd., Jiangmen, China) for about 6 weeks. Before the trial, the fish (initial weight: 9.71 ± 0.04 g) were gathered up. For each replicate, 30 fish were randomly selected to be anesthetized with 0.01% 2-phenoxyethanol and weighed; three replicates were established in each group. In the 65-day rearing trial, golden pompano were fed twice daily at 6:00 a.m. and 5:00 p.m. until they did not scramble at the water surface; the seawater temperature was 24–30 °C and salinity was 28–31% during the trial.

### 2.3. Sample Collection

Before sampling, the fish in each experimental group were fasted for 24 h. In each net cage, four fish were caught at random for anesthesia; subsequently, serum and muscle tissue samples (12 samples per group) were collected and preserved in liquid nitrogen. Another four fish were caught from each net (12 fish per group) for the determination of muscle textural properties and body composition. Finally, all the remaining fish in the nets were retrieved, counted, and weighed.

### 2.4. Proximate Composition and Serum Biochemical Index Assays

The crude protein, crude lipid, ash, and moisture content of the diets, whole fish, and tissue in this experiment were referred to the standard methods of the Association of Official Analytical Chemists, and a specific determination procedure was carried out with reference to that described by Ma et al. [32]. Of these, feed samples were measured in three replicates per group, while whole fish and tissue samples were measured in six replicates per group.

Among the biochemical indexes of serum in this experiment, the indexes related to protein metabolism, such as TP, ALB, and BUN content, and the indexes related to lipid metabolism, such as TG, T-CHO, HDL cholesterol, and LDL cholesterol, were determined by assay kits (Nanjing Jiancheng Bioengineering Co., Ltd., Nanjing, China) using a microplate reader (BilTek Instruments, Inc., Winooski, VT, USA), and six samples of each group were measured as replicates for the above indexes.

### 2.5. Amino Acid and Fatty Acid Composition Assays

The dietary amino acid composition in this experiment was assayed according to the acid hydrolysis method [34]. In brief, first, 10 mL of 6 mol/L HCl was added to the sample (6 replicates per group) and hydrolyzed at 110 °C for 22 h. Subsequently, the samples were filtered through filter paper and dried in a water bath (65 °C); the residual HCl was eluted with double-distilled water, and the procedure was repeated twice. Finally, 4 mL of the buffer (pH = 2.2) was added, and the samples were loaded into the injection vials and assessed with an analysis system (L-8900, Hitachi, Tokyo, Japan).

The diets and tissue fatty acid composition in this experiment were assayed with reference to previously published methods of our research group [32]. As replicates, three feed samples and six tissue samples per group were measured after extracting the lipid from the samples by soaking them in the chloroform/methanol (2:1) solution for 24 h in centrifuge tube I. Then, the chloroform layer was mixed well with distilled water and aspirated into centrifuge tube II, and the crude lipid samples were blown dry with N_2_ in a water bath (45 °C). The samples were saponified by adding the 0.5 M KOH–methanol solution into centrifuge tube II and shaking in a water bath (65 °C) for 1 h. Then the centrifuge tube II was added with anhydrous methanol and boron trifluoride methanol solution, and subsequently placed in a water bath (72 °C) for 45 min. Hexane and saturated saline were added to dissolve fatty acid methyl esters and centrifuged at 12,000 rpm, then 500 µL of supernatant was transferred to a brown injection bottle and assessed using gas chromatography (7890B, Agilent, Palo Alto, CA, USA).

### 2.6. Textural Properties Assays

The test method for the textural properties of muscle was performed with reference to the operational procedure described by Ma et al. [32], in which six samples per group were measured as replicates. Firstly, thin slices of uniform width and thickness (approximately 2.0 cm × 2.0 cm × 0.5 cm) were cut out of the fresh fish dorsal muscle (6 samples per group). Subsequently, all samples were assessed with a texture analyzer (Shanghai turnkey pull, Shanghai, China).

### 2.7. Real-Time Quantitative PCR Assay

Firstly, the total RNA extraction kit (BioFlux, Beijing, China) was used to extract the total RNA from the muscle (6 replicates per group). All samples were diluted to achieve the same RNA concentration. RNA reverse transcription was carried out using the PrimeScript TMRT reagent kit (Takara, Tokyo, Japan). The primers used in this experiment are shown in Table 4. The qRT-PCR was assayed by SYBR^®^ Green Master Mix (Toyobo Co., Ltd., Osaka, Japan) with a CFX Connect Real-Time System (Bio-Rad Laboratories, Inc., Hercules, CA, USA), which was previously described by Zhang et al. [33]. Finally, statistical analysis of all data was carried out with the 2^−ΔΔCT^ method.

### 2.8. Estimation and Statistical

The following formulas were used to estimate the below indicators:Weight gain (WG, g) = final body weight (g) − initial body weight (g)
Weight gain rate (WGR, %) = 100 × [final body weight (g) − initial body weight (g)/initial body weight (g)]
Specific growth rate (SGR, %/day) = 100 × [ ln final weight (g) − ln initial weight (g)]/days of feeding trial
Feed conversion ratio (FCR) = dry feed consumed (g)/wet weight body gain (g)
Survival rate (SR, %) = 100 × (final fish number/initial fish number)
Viscerosomatic index (VSI, %) = 100 × [viscera wet weight (g)/final body weight (g)]
Hepatosomatic index (HSI, %) = 100 × [liver wet weight (g)/final body weight (g)]
Condition factor (CF, g/cm^−3^) = 100 × [final body weight (g)/body length (cm)^3^]

The data of this trial were expressed as mean ± SEM. One-way ANOVA was used to analyze the data in this trial with SPSS software (Ver 22.0, International Business Machines Co., Ltd., Armonk, NY, USA). The Tukey’s multiple comparison method was utilized to analyze the significance of differences between groups, and *p* < 0.05 means that the data was significantly different.

## 3. Results

### 3.1. Growth and Somatic Indexes

Table 5 shows that the survival of each group was 100%, and the CF was not significantly different among all groups. Compared to the Control group (24% FM), the WG, WGR, SGR, and FCR of the 25NGP group (18% FM) showed no significant difference, whereas the WG, WGR, and SGR of the 50NGP (12% FM) group and the 75NGP (6% FM) group significantly decreased and the FCR significantly increased (*p* < 0.05). Moreover, the VSI and HSI of the 25NGP–75NGP groups were not significantly different from those of the Control group. Furthermore, the VSI of 25NGP was significantly decreased compared to that of the 50NGP and 75NGP groups, while the HSI of 25NGP showed a significant decrease compared to that of the 75NGP group (*p* < 0.05). The above results indicate that the substitution of 25% of dietary FM with NGCP is feasible, without affecting the growth and somatic indexes of golden pompano.

### 3.2. Whole-Body and Muscle Proximate Composition

Table 6 shows that there was no significant difference in all indexes of whole-body and muscle proximate composition between the 25NGP and Control groups. However, compared to the Control group, the muscle crude lipid content in the 75NGP group was significantly decreased, and the whole-body ash content and muscle moisture content were significantly increased (*p* < 0.05). Besides, the whole-body ash content of the 25NGP group was significantly decreased compared to the 75NGP group, while the muscle crude lipid content of the 25NGP group was significantly increased compared to the 50NGP and 75NGP groups (*p* < 0.05). The above results indicate that the substitution of 25% FM with NGCP had no negative impacts on the proximate composition of golden pompano.

### 3.3. Serum Biochemical Indexes

Figure 1 showed that, compared to the Control group, the TP, ALB, and BUN levels of the 25NGP group had no significant difference, whereas the TP and ALB levels of the 50NGP and 75NGP groups were significantly higher (*p* ˂ 0.05). In addition, the TP level of the 25NGP group was significantly lower than the 75NGP group, and the ALB level of the 25NGP group was significantly lower than the 50NGP and 75NGP groups (*p* ˂ 0.05).

In serum lipid metabolism indexes, compared to the Control group, the T-CHO and TG levels of the 25NGP group were not significantly different, but T-CHO levels in the 50NGP group, TG levels in group 75NGP, and LDL-C levels in groups 25NGP and 50NGP were significantly higher (*p* ˂ 0.05). The HDL-C levels in each group had no significant difference.

### 3.4. Textural Properties

The muscle textural properties of the juvenile golden pompano are shown in Figure 2. The muscle hardness, chewiness, and gumminess in each group showed no significant difference. Compared to the Control group, the tenderness, springiness, and cohesiveness of the 25NGP group were not significantly different, whereas the tenderness of the 75NGP group was significantly decreased. Furthermore, the tenderness of the 25NGP group was significantly higher than that of the 50NGP and 75NGP groups (*p* < 0.05). The above results indicate that the substitution of 25% FM with NGCP had no negative impact on the muscle textural properties of golden pompano.

### 3.5. Fatty Acid Composition in Muscle

Table 7 shows that, compared to the Control group, the muscle n-3 PUFA content in the 25NGP-75NGP groups was significantly decreased, and the 75NGP group was significantly decreased compared to the 25NGP-50NGP groups (*p* < 0.05). The muscle n-6 PUFA content of the 25NGP-50NGP groups was not significantly different from the Control group, whereas the 75NGP group was significantly decreased (*p* < 0.05). In addition, the muscle SFA and MUFA content in each group showed no significant differences.

### 3.6. Protein Metabolism in Muscle

Figure 3 shows that, compared to the Control group, the mRNA expression of *4e-bp1* in the 25NGP and 50NGP groups had no significant difference, but the 75NGP group was significantly up-regulated (*p* < 0.05). Moreover, the mRNA expression of *mtor* and *s6k1* in each group had no significant differences. The above results indicate that the substitution of 50% FM with NGCP did not negatively affect the protein metabolism of golden pompano.

### 3.7. Lipid Metabolism in Muscle

Figure 4 showed that, compared to the Control group, the mRNA expression of *pparγ* in the 25NGP–50NGP groups had no significant difference, but the 75NGP group was significantly down-regulated (*p* < 0.05). Moreover, there was no significant difference in the mRNA expression of *fas*, *srebp1*, and *acc1* among all groups. The above results indicate that the substitution of 50% FM with NGCP had no negative effects on the lipid metabolism of juvenile golden pompano.

## 4. Discussion

Those inexpensive and widely available animal and plant processing by-products should be considered more to replace FM and soybean meal in aquafeeds. Previously, spray-dried blood cell meal was proven to be a viable alternative to FM in the diet of whiteleg shrimp (*Litopenaeus vannamei*), but the percentage of substitution should not be higher than 60%; otherwise, the growth performance was reduced [36]. Moreover, a study on ussuri catfish (*Pseudobagrus ussuriensis*) found that growth performance was significantly reduced, while 40% of dietary FM was replaced by meat and bone meal [37]. These studies suggest that the replacement of high percentages of FM in aquafeeds by a particular feed protein is difficult because it affects the growth of aquatic animals. In the present study, bovine bone meal, dephenolized cottonseed protein, and blood cell meal were used to produce an NGCP to replace FM in the feed of golden pompano. After 65 days of culture trials, the survival rate of all groups was found to be 100%, indicating that all feeds were safe for fish. Furthermore, the growth indexes of fish in the 25NGP group were not affected when 25% dietary FM was substituted with NGCP, which proved that the incorporation of non-grain proteins in golden pompano feed was feasible. These findings were comparable to previous reports on the replacement of FM by compound proteins in aquafeeds; for example, a study performed on gibel carp (*Carassius auratus gibelio*) found that the entire FM in the feed could be replaced by compound plant protein without reducing growth performance [38]. Similarly, the replacement of dietary FM with a mixture of shrimp hydrolysate and plant proteins did not diminish the growth of largemouth bass (*Micropterus salmoides*), and the weight gain reached its maximum at the replacement ratio of 22.2% [39]. These studies illustrated that the application of non-grain mixed proteins in fish feeds was superior compared to those containing single proteins. Regarding growth indicators as a reference, the dietary FM of golden pompano could be reduced to 18% by NGCP substitution.

The muscle quality and serum indexes are also important aspects in assessing the feasibility of NGCP application in feeds. In terms of muscle quality, first, the textural properties of muscle determine the physical properties and affect the taste of food [40]. The results of this study showed that 25% and 50% FM replacement by NGCP had no negative effect on muscle tenderness, hardness, springiness, chewiness, gumminess, or cohesiveness in fish. Moreover, the proximate composition of whole fish and muscle signifies the edibility of fish products [41]. In this study, the substitution of 25% and 50% dietary FM with NGCP had no negative impact on the whole-body or muscle proximate composition. Therefore, regarding nutritional value, it is feasible to reduce the proportion of FM in golden pompano feed to at least 12% using NGCP as a replacement. On the other hand, serum biochemical indexes are also used to assess the health and nutritional metabolism of fish [42]. Serum TP—comprising ALB and GLB—has several physiological roles, such as maintaining the osmotic pressure and pH of blood vessels, transporting metabolites of the body, regulating transported substances, and being an important indicator of feed protein absorption and metabolism [43]. Serum HDL-C and LDL-C are responsible for the transport of cholesterol between the liver and blood or tissues [44,45], and the levels of TG and T-CHO are closely associated with excessive liver lipid deposition [46]. The present findings indicate that serum TP, ALB, TG, T-CHO, and HDL-C showed no significant differences between the group with 25% dietary FM replacement with NGCP and the control group, indicating that this replacement ratio is feasible. The above-mentioned muscle quality and serum biochemical indexes signify the potential and applicability of NGCP in golden pompano feed.

However, there are limitations to the incorporation of NGCP in golden pompano feed. In this study, the substitution of more than 50% dietary FM with NGCP significantly decreased certain indicators of growth performance, muscle quality, and serum biochemical indexes of juvenile fish, which may be related to the following factors.

The content of limiting amino acids (LAA, lysine, and methionine) markedly decreased in the groups with high replacement ratios of dietary FM. It is well known that amino acids can be classified as essential amino acids (EAA) and non-essential amino acids (NEAA), of which EAA are those that are insufficiently synthesized or cannot be synthesized by animals, and they must be provided in the diet to ensure the normal growth of farmed animals [47]. In aquatic animals, lysine [48], methionine [49], and arginine [50] are considered LAA, and their deficiency affects the growth of fish. The dietary amino acid composition in this study showed that lysine and methionine decreased as the ratio of dietary FM replacement by NGCP increased. It has been reported that the levels of dietary arginine [51] and lysine [52] affect the expression of the target of rapamycin (TOR) pathway, which is believed to respond to nutritional status in fish, leading to protein synthesis and growth [53]. The eIF4E-binding protein and ribosomal protein S6 kinase 1 are downstream effectors in the TOR pathway that regulate mTOR activity in an antagonistic manner [54]. In this study, the expression level of *4e-bp1* in muscle was significantly up-regulated when 75% of dietary FM was replaced by NGCP. The study on largemouth bass showed that adding protein hydrolysate to the diet improved the weight gain of fish and activated targets of the TOR pathway, including upregulation of tor and akt1 and downregulation of *4e-bp1* [55]. Similarly, growth performance showed an opposite trend to the expression level of *4e-bp1* in the study of grass carp (*Ctenopharyngodon idella*) [56]. Therefore, the increased expression of *4e-bp1*, an antagonistic factor, may be responsible for the inhibition of protein synthesis in the muscles of the golden pompano, which may contribute to the growth performance results described above.

In this study, diets with low FM showed an imbalance of fatty acid composition; the SFA and MUFA levels increased and the n-3 PUFA levels decreased with the decrease in the dietary FM content. In most marine carnivorous fish, the ability to convert linoleic acid (LA) and linolenic acid (LNA) to highly unsaturated fatty acids (HUFAs) is poor [57]. The golden pompano, a marine carnivorous fish, has a high dietary requirement for n-3 HUFAs to maintain normal growth and metabolism [58]. A deficiency of n-3 HUFAs may lead to abnormalities in physiological functions. In general, the fatty acid deposition of tissue is consistent with dietary fatty acids [59]. Similarly, in this study, the muscle fatty acid composition signified the fatty acid composition of diets, where the SFA content of muscle increased with the proportion of FM substitution, whereas the n-3 PUFA content subsequently decreased. Therefore, the impaired growth of golden pompano mentioned above may also be due to an imbalance of dietary fatty acids. In addition, the imbalance of dietary fatty acids led to the abnormal metabolism of lipid deposition in fish. Several studies have shown that polyunsaturated fatty acids facilitate the activation of PPARγ ligands and promote lipid synthesis [60,61]. PPAR is a ligand-activated receptor in the nuclear hormone receptor family, including PPARα, PPARβ, and PPARγ, which control many cellular metabolic processes, among which PPARγ is primarily involved in the regulation of lipid storage [62]. In this study, the expression levels of pparγ were significantly down-regulated after 75% replacement of dietary FM. This may be due to the decrease in muscle crude lipid when a high percentage of dietary FM was replaced. Similarly, a study performed on grass carp showed that the crude lipid content of muscle was positively related to the expression of *pparγ* [63]. Moreover, it has been reported that the textural properties of fish may be influenced by the oil source and fatty acid composition of the feed [64]. It has been reported that a higher content of SFAs than unsaturated fatty acids in muscle increases muscle hardness [65]. In addition, the textural properties are affected by the crude lipid content of muscle; for example, muscle hardness tends to increase when the muscle lipid content is low [66]. In this study, muscle tenderness and crude lipid content showed significant reductions when 75% of the dietary FM was replaced by NGCP. This finding was contrary to the previous findings, which may be due to other factors used in this experiment, such as plant protein. A study in Senegalese sole (*Solea senegalensis*) showed that 100% substitution of dietary FM by mixed plant protein decreased the textural properties of muscle [67], indicating that the addition of plant protein to the feed has a significant effect on the textural properties of fish muscles, and similar findings have been reported in grass carp [68]. In addition, it has been reported that feed lysine levels affect muscle hardness by mediating the development of muscle fibers [69]. Therefore, the low lysine content in plant protein may also be the reason for the reduced muscle quality when a high percentage of FM is substituted. The findings indicate that when the dietary FM was replaced by a high proportion of NGCP, the resulting low n-3 PUFA content in the diet may have led to the above-mentioned decrease in muscle quality, whereas the differences in textural properties may be attributed to certain components of plant proteins, which need to be further analyzed and explored.

## 5. Conclusions

The present study found that dephenolized cottonseed protein, bovine bone meal, and blood cell meal could be combined to form a compound protein to replace at least 25% of FM in the feed without reducing growth performance or muscle quality, so that the proportion of FM in the diet of golden pompano could be reduced to 18%. However, the imbalance of amino acids and fatty acids in feed caused by high ratio substitution (higher than 50%) might affect the protein and lipid metabolism of muscle. This study provides new ideas to alleviate the shortage of protein sources in aquafeeds and broaden the application of non-grain proteins.

## Figures and Tables

**Figure 1 animals-13-01704-f001:**
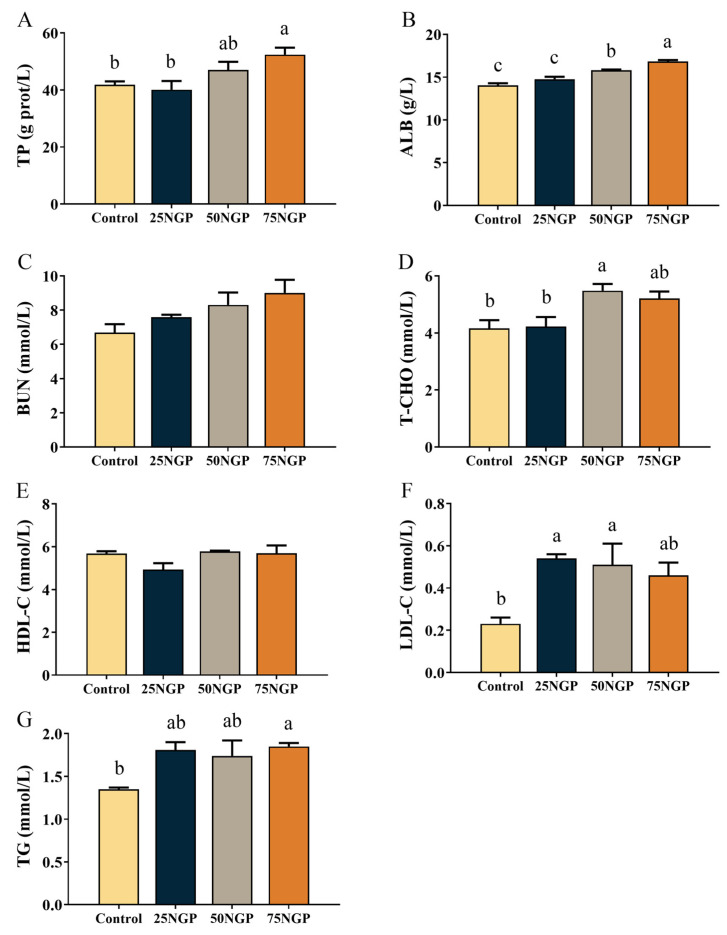
Serum biochemical indexes of juvenile golden pompano fed different diets for 65 days. Values are presented as mean ± SEM (n = 6). Significance analysis between groups was performed using Tukey’s multiple comparison method, and bars without sharing a common letter indicate a significant difference (*p* < 0.05), while those lacking letters indicate no significant difference. (**A**): TP, total protein (g L^−1^); (**B**): ALB, albumin (g L^−1^); (**C**): BUN, blood urea nitrogen (mmol L^−1^); (**D**): T-CHO, total cholesterol (mmol L^−1^); (**E**): HDL-C, high-density lipoprotein cholesterol (mmol L^−1^); (**F**): LDL-C, low-density lipoprotein cholesterol (mmol L^−1^); (**G**): TG, triglyceride (mmol L^−1^).

**Figure 2 animals-13-01704-f002:**
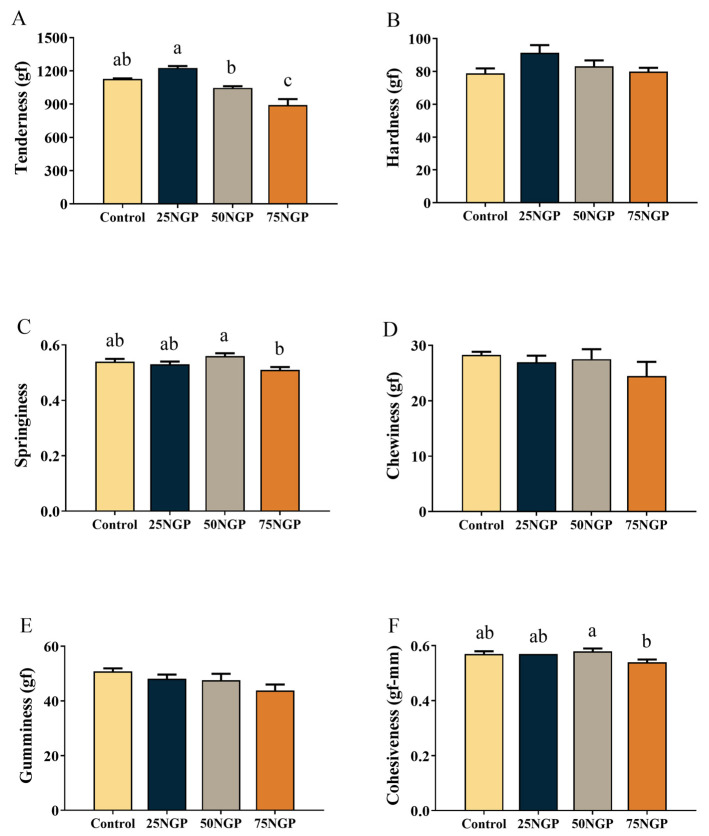
Textural properties indexes in the muscle of juvenile golden pompano fed different diets for 65 days. Values are presented as mean ± SEM (n = 6). Significance analysis between groups was performed using Tukey’s multiple comparison method, and bars without sharing a common letter indicate a significant difference (*p* < 0.05), while those lacking letters indicate no significant difference. (**A**): Tenderness (gf); (**B**): Hardness (gf); (**C**): Springiness; (**D**): Chewiness (gf); (**E**): Gumminess (gf); (**F**): Cohesiveness (gf-mm).

**Figure 3 animals-13-01704-f003:**
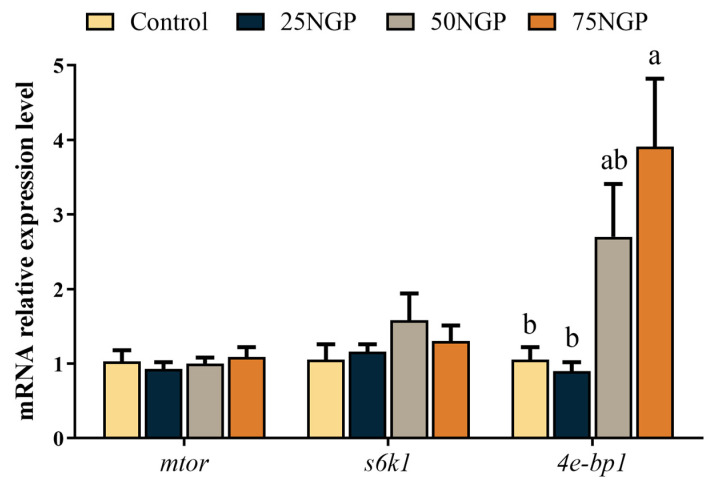
mRNA expression levels of genes related to the protein metabolism of juvenile golden pompano fed different diets for 65 days. Values are presented as mean ± SEM (n = 6). Significance analysis between groups was performed using Tukey’s multiple comparison method, and the bars without sharing a common letter indicate a significant difference (*p* < 0.05), while those lacking letters indicate no significant difference. *mtor*, mechanistic target of rapamycin kinase; *s6k1*, ribosomal S6 kinase; *4e-bp1*, 4E-binding protein 1.

**Figure 4 animals-13-01704-f004:**
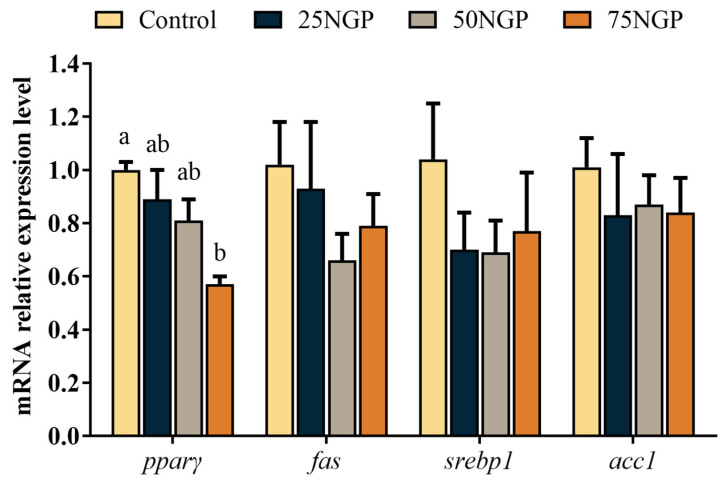
mRNA expression levels of genes related to the lipid metabolism of juvenile golden pompano fed different diets for 65 days. Values are presented as the mean ± SEM (n = 6). Significance analysis between groups was performed using Tukey’s multiple comparison method, and the bars with different letters are significantly different (*p* < 0.05). *pparγ*, peroxisome proliferator-activated receptor γ; *fas*, fatty acid synthase; *srebp1*, sterol regulatory element binding proteins 1; *acc1*, acetyl coenzyme a carboxylase 1.

**Table 1 animals-13-01704-t001:** Ingredients and nutrient composition of the experimental diets (% dry weight).

	Diets			
	Control	25NGP	50NGP	75NGP
Percentage of FM replaced by NGCP	0	25	50	75
Fishmeal	24.00	18.00	12.00	6.00
Non-grain compound protein ^1^		6.00	13.00	19.50
Soybean protein concentrate	7.00	7.00	7.00	7.00
Poultry meal	8.00	8.00	8.00	8.00
Corn gluten meal	9.00	9.00	9.00	9.00
Fermented soybean meal	10.00	10.00	10.00	10.00
Compound oil ^2^	8.00	8.10	8.40	8.70
Flour	17.00	17.00	17.00	17.00
Vitamin mineral mixture ^3^	1.00	1.00	1.00	1.00
Premix compound ^4^	2.50	2.50	2.50	2.50
Corn husk	13.50	13.27	11.86	10.94
L-lysine		0.07	0.12	0.18
DL-methionine		0.06	0.12	0.18
Total	100.00	100.00	100.00	100.00
Proximate composition (% dry weight)				
Dry weight	92.34	91.47	92.21	91.97
Crude protein	45.13	44.96	45.19	45.20
Crude lipid	12.18	11.98	11.96	11.95
Ash	8.81	8.68	8.57	8.62

Fishmeal, bovine bone meal, dephenolized cottonseed protein, blood cell meal, soybean protein concentrate, poultry meal, corn gluten meal, fermented soybean meal, flour, corn husk, L-lysine, DL-methionine, and premix compound were provided by Taishan Xiangxing Feed Co., Ltd., Taishan, China. ^1^ NGCP consists of 50% dephenolized cottonseed protein, 30% bovine bone meal, and 20% blood cell meal. ^2^ Consists of fish, soybean, rapeseed, perilla, palm oils, and phospholipids, together with small amount of emulsifier and antioxidant, which were provided by Guangzhou UBT Feed Technology Co., Ltd., Guangzhou, China. Details are not shown here due to the application for a patent. ^3^ Consists of a vitamin mixture and mineral compound. Vitamin mixture (per kg): VA: 1100000 IU; VD3: 320000 IU; VK3: 1000 mg; VB1: 1500 mg; VB2: 2800 mg; VC: 17 mg; VE: 8 mg; calcium pantothenate: 2000 mg; nicotinamide: 7800 mg; folic acid: 400 mg; inositol: 12800 mg; VB6: 1000 mg. Mineral compound (per kg): potassium iodide: 0.8 mg; sodium fluoride: 2 mg; copper sulfate: 10 mg; cobalt chloride (1%): 50 mg; zinc sulfate: 50 mg; calcium sulfate: 80 mg; magnesium sulfate: 1200 mg; manganese sulfate: 60 mg; table salt: 100 mg; zeolite powder: 15.45 g. Purchased from Guangdong Yuehai Feeds Group Co., Ltd., Zhanjiang, China. ^4^ Consists of choline chloride, monocalcium phosphate, ethoxyquinoline, and glycine betaine.

**Table 2 animals-13-01704-t002:** Amino acid composition of the experimental diets (% dry weight).

Amino Acids	Control	25NGP	50NGP	75NGP
Lys	2.27	2.25	2.16	2.09
Phe	1.90	1.96	2.03	2.10
Met	0.77	0.74	0.72	0.71
Thr	1.72	1.71	1.65	1.61
Ile	1.64	1.49	1.50	1.37
Leu	3.71	3.71	3.78	3.75
Val	1.95	1.87	2.00	1.98
Arg	2.37	2.54	2.69	2.81
His	1.10	1.11	1.12	1.13
EAA	17.43	17.38	17.65	17.55
Asp	3.54	3.62	3.62	3.70
Ser	1.94	2.01	1.98	2.05
Glu	7.39	7.56	7.62	7.84
Gly	2.18	2.22	2.23	2.32
Ala	2.75	2.73	2.73	2.75
Cys	0.49	0.47	0.49	0.49
Pro	2.76	2.76	2.80	2.86
NEAA	21.05	21.37	21.47	22.01

Data are the mean of three duplicate determinations. EAA: Essential amino acids, including Lys, Phe, Met, Thr, Ile, Leu, Val, Arg, and His. NEAA: Non-essential amino acids, including Asp, Ser, Glu, Gly, Ala, Cys, and Pro.

**Table 3 animals-13-01704-t003:** Fatty acid composition of the experimental diets (% total fatty acids).

Main Fatty Acids	Control	25NGP	50NGP	75NGP
14:0	5.51	5.72	6.09	5.88
15:0	0.47	0.51	0.55	0.57
16:0	24.94	26.61	27.15	29.90
17:0	0.68	0.73	0.71	0.75
18:0	4.11	4.53	4.44	5.42
20:0	0.21	0.23	0.34	0.36
SFA	35.93	38.26	39.16	42.88
16:1	5.66	5.62	5.71	5.73
17:1	0.86	0.91	0.95	0.85
18:1n-9	20.78	21.75	22.02	24.79
MUFA	27.31	28.29	28.68	31.38
18:2n-6	16.78	17.33	17.49	15.86
18:3n-6	1.43	1.41	1.41	1.08
20:3n-6	0.20	0.56	0.51	0.30
20:4n-6	0.73	0.80	0.73	0.75
n-6 PUFA	19.65	20.10	20.13	17.99
18:3n-3	0.99	0.95	1.05	0.65
20:5n-3	6.08	4.43	3.79	1.85
22:6n-3	5.43	4.14	3.50	1.58
n-3 PUFA	12.49	9.52	8.34	4.08
n-3/n-6	0.64	0.47	0.41	0.23

Data are the mean of three duplicate determinations. SFA: saturated fatty acids, including 14:0, 15:0, 16:0, 17:0, 18:0, and 20:0. MUFA: monounsaturated fatty acids, including 16:1, 17:1, and 18:1n-9. n-6 PUFA: n-6 polyunsaturated fatty acids, including 18:2n-6 (LA), 18:3n-6, 20:3n-6, and 20:4n-6 (ARA). n-3 PUFA: n-3 polyunsaturated fatty acids, including 18:3n-3 (ALA), 20:5n-3 (EPA), and 22:6n-3 (DHA).

**Table 4 animals-13-01704-t004:** Nucleotide sequences of the primers used to assay gene expressions by real-time PCR.

Target Gene	Forward Primer (5′-3′)	Reverse Primer (3′-5′)	Reference
*mtor*	GATCAGGAGAGAGGCCATCC	AGCCGGGTAAAACTCATCCA	Genome sequences
*s6k1*	GAAGCCCAAGAACACCTGTG	GCTTGTGTCCATTTGCTCCA	Genome sequences
*4e-bp1*	GGGACTCTGTTCAGCACCA	CGGTTGAGTCACTGGGTTTG	Genome sequences
*pparγ*	TCAGGGTTTCACTATGGCGT	CTGGAAGCGACAGTATTGGC	Genome sequences
*fas*	GATGGATACAAAGAGCAAGG	GTGGAGCCGATAAGAAGA	Genome sequences
*srebp1*	GAGCCAAGACAGAGGAGTGT	GTCCTCTTGTCTCCCAGCTT	Genome sequences
*acc1*	GTGGAGCCGATAAGAAGA	GCTTCCAGCAGCAAACG	Genome sequences
*β-actin*	TACGAGCTGCCTGACGGACA	GGCTGTGATCTCCTTCTGC	Tan et al. [35]

The primers used in this experiment, according to Tan et al. [35], and the genome sequences of golden pompano (10.6084/m9.figshare.7570727.v3). *mtor*, mechanistic target of rapamycin kinase; *s6k1*, S6 kinase 1; *4e-bp1*, 4E binding protein 1; *pparγ*, peroxisome proliferators-activated receptor gamma; *fas*, fatty acid synthase; *srebp1*, sterol regulatory element binding protein 1; *acc1*, acetyl-CoA-carboxylase.

**Table 5 animals-13-01704-t005:** Growth performance, feed utilization, and morphometric parameters of juvenile golden pompano fed different diets for 65 days.

Groups	Control	25NGP	50NGP	75NGP
Initial body weight	9.72 ± 0.11	9.61 ± 0.15	9.72 ± 0.11	9.78 ± 0.14
Final body weight	86.82 ± 2.47 ^a^	80.12 ± 0.65 ^b^	74.24 ± 1.00 ^b^	61.11 ± 0.92 ^c^
Weight gain	77.1 ± 2.46 ^a^	70.51 ± 0.65 ^ab^	64.52 ± 1.11 ^b^	51.33 ± 0.90 ^c^
Weight gain rate	793.43 ± 26.89 ^a^	733.98 ± 14.00 ^ab^	655.72 ± 23.53 ^b^	524.83 ± 11.11 ^c^
Specific growth rate	3.36 ± 0.05 ^a^	3.26 ± 0.03 ^ab^	3.13 ± 0.04 ^b^	2.82 ± 0.03 ^c^
Feed conversion ratio	1.17 ± 0.05 ^c^	1.25 ± 0.01 ^bc^	1.40 ± 0.03 ^b^	1.73 ± 0.05 ^a^
Survival rate	100.00 ± 0.00	100.00 ± 0.00	100.00 ± 0.00	100.00 ± 0.00
Viscerosomatic index	6.22 ± 0.09 ^ab^	5.83 ± 0.24 ^b^	6.60 ± 0.19 ^a^	6.76 ± 0.18 ^a^
Hepatosomatic index	1.45 ± 0.11 ^ab^	1.08 ± 0.09 ^b^	1.32 ± 0.12 ^ab^	1.54 ± 0.08 ^a^
Condition factor	3.02 ± 0.08	3.14 ± 0.08	3.11 ± 0.07	3.23 ± 0.13

The results of initial body weight (IBW, g fish^−1^), final body weight (FBW, g fish^−1^), weight gain (WG, g), weight gain rate (WGR, %), specific growth rate (SGR, % day^−1^), feed conversion ratio (FCR), and survival rate (SR, %) are presented as the mean ± SEM (n = 3), and the results of viscerosomatic index (VSI, %), hepatosomatic index (HSI, %), and condition factor (CF, g cm^−3^) are presented as the mean ± SEM (n = 6). Significance analysis between groups was performed using Tukey’s multiple comparison method, and in each row, means without sharing a common letter are significantly different (*p* < 0.05), while those lacking letters indicate no significant difference.

**Table 6 animals-13-01704-t006:** Whole body and muscle composition (% dry weight) of the juvenile golden pompano fed different diets for 65 days.

Groups	Control	25NGP	50NGP	75NGP
Whole body				
Crude protein	54.17 ± 0.28	56.42 ± 0.98	57.16 ± 0.19	56.05 ± 1.21
Crude lipid	31.3 ± 0.98	29.27 ± 1.42	31.12 ± 0.19	29.46 ± 0.52
Ash	12.2 ± 0.19 ^b^	12.51 ± 0.34 ^b^	12.62 ± 0.12 ^b^	13.84 ± 0.29 ^a^
Moisture	66.49 ± 0.41	68.14 ± 0.74	67.92 ± 0.15	67.82 ± 0.4
Muscle				
Crude protein	83.36 ± 0.94	82.12 ± 0.75	85.18 ± 0.61	85.09 ± 0.9
Crude lipid	9.84 ± 0.55 ^ab^	11.44 ± 0.95 ^a^	6.09 ± 1.21 ^bc^	5.98 ± 0.45 ^c^
Ash	6.23 ± 0.13	5.84 ± 0.17	6.11 ± 0.25	5.99 ± 0.29
Moisture	74.79 ± 0.27 ^b^	75.09 ± 0.34 ^ab^	75.77 ± 0.26 ^ab^	75.93 ± 0.07 ^a^

Results are presented as the mean ± SEM (n = 6). Significance analysis between groups was performed using Tukey’s multiple comparison method, and in each row, means without sharing a common letter are significantly different (*p* < 0.05), while those lacking letters indicate no significant difference.

**Table 7 animals-13-01704-t007:** Fatty acid composition (% total fatty acids) in the muscle of juvenile golden pompano fed different diets for 65 days.

Main Fatty Acids	Control	25NGP	50NGP	75NGP
14:0	3.73 ± 0.23	3.63 ± 0.09	3.53 ± 0.11	3.59 ± 0.05
15:0	0.39 ± 0.01 ^b^	0.43 ± 0.01 ^a^	0.42 ± 0.00 ^a^	0.43 ± 0.01 ^a^
16:0	26.04 ± 0.30 ^b^	27.3 ± 0.13 ^b^	26.83 ± 0.48 ^b^	30.03 ± 0.28 ^a^
17:0	0.49 ± 0.01 ^c^	0.51 ± 0.01 ^bc^	0.54 ± 0.01 ^a^	0.52 ± 0.00 ^ab^
18:0	5.09 ± 0.08 ^c^	5.76 ± 0.04 ^b^	6.55 ± 0.06 ^a^	6.58 ± 0.11 ^a^
20:0	0.23 ± 0.01	0.25 ± 0.01	0.38 ± 0.08	0.44 ± 0.06
SFA	35.1 ± 1.24	37.06 ± 0.81	36.03 ± 1.24	37.99 ± 1.24
16:1	4.98 ± 0.10 ^a^	4.75 ± 0.06 ^ab^	4.51 ± 0.05 ^b^	4.68 ± 0.03 ^b^
17:1	0.71 ± 0.01 ^a^	0.68 ± 0.02 ^ab^	0.62 ± 0.02 ^b^	0.64 ± 0.02 ^b^
18:1n-9	25.11 ± 0.13 ^c^	26.51 ± 0.22 ^b^	26.23 ± 0.23 ^b^	29.36 ± 0.21 ^a^
20:1	1.73 ± 0.13	1.67 ± 0.02	1.64 ± 0.04	1.86 ± 0.03
MUFA	32.36 ± 0.44	33.61 ± 0.20	30.78 ± 1.54	34.71 ± 0.93
18:2n-6	11.94 ± 0.15 ^bc^	12.44 ± 0.14 ^ab^	12.64 ± 0.19 ^a^	11.3 ± 0.12 ^c^
20:3n-6	0.77 ± 0.04 ^a^	0.76 ± 0.06 ^a^	0.53 ± 0.03 ^b^	0.28 ± 0.03 ^c^
20:4n-6	0.5 ± 0.08 ^a^	0.59 ± 0.03 ^a^	0.26 ± 0.01 ^b^	0.21 ± 0.01 ^b^
n-6 PUFA	12.99 ± 0.27 ^a^	13.5 ± 0.37 ^a^	13.26 ± 0.26 ^a^	11.78 ± 0.07 ^b^
18:3n-3	0.28 ± 0.01 ^a^	0.28 ± 0.01 ^a^	0.21 ± 0.00 ^b^	0.13 ± 0.01 ^c^
20:5n-3	1.93 ± 0.05 ^a^	1.47 ± 0.02 ^b^	1.22 ± 0.05 ^c^	0.72 ± 0.02 ^d^
22:6n-3	6.76 ± 0.09 ^a^	5.77 ± 0.04 ^c^	6.27 ± 0.10 ^b^	3.13 ± 0.08 ^d^
n-3 PUFA	8.9 ± 0.14 ^a^	7.53 ± 0.05 ^b^	7.67 ± 0.19 ^b^	3.86 ± 0.04 ^c^
n-3/n-6	0.67 ± 0.00 ^a^	0.55 ± 0.02 ^b^	0.58 ± 0.04 ^b^	0.33 ± 0.00 ^c^

Results are presented as the mean ± SEM (n = 6). Significance analysis between groups was performed using Tukey’s multiple comparison method, and in each row, means without sharing a common letter are significantly different (*p* < 0.05), while those lacking letters indicate no significant difference. SFA: saturated fatty acids, including 14:0, 15:0, 16:0, 17:0, 18:0, and 20:0. MUFA: monounsaturated fatty acids, including 16:1, 17:1, 18:1n-9, and 20:1. N-6 PUFA: n-6 polyunsaturated fatty acids, including 18:2n-6, 20:3n-6, and 20:4n-6. N-3 PUFA: n-3 polyunsaturated fatty acids, including 18:3n-3, 20:5n-3, and 22:6n-3.

## Data Availability

All data are contained within the article.

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
