# Peer review of "Dietary Fishmeal Can Be Partially Replaced with Non-Grain Compound Proteins through Evaluating the Growth, Biochemical Indexes, and Muscle Quality in Marine Teleost Trachinotus ovatus"

_animals, 2023, doi:10.3390/ani13101704_

Round 1
Reviewer 1 Report
The authors investigated the effect of Dietary fishmeal can be partially replaced with non-grain compound proteins by evaluating the growth, biochemical indexes, and muscle quality in marine teleost silverfish Trachinotus ovatus. They designed four treatments to test their hypothesis. This manuscript (MS) was clearly written and easy to understand. They covered a wide range of physical factors, from growth to gene expression. This work can help the sustainability of this species farming as few studies have been done on this topic. However, some major issues significantly compromised the quality of this MS.
Major comments:
- First, the manuscript needs to be edited by a native English speaker to improve the language of the MS and fix errors. A deep revision is required.
Other comments:
Line 13 , revise~~!!
· Line 16, mention them; what are they?
· Line 18, common name.
· Line 26, common name and scientific name.
· Line 31, is not correct for the final weight and I think weight gain. Please add weight gain data as well. The weight gain (g) is the most important parameter and if there is a significant difference, just report that this fish needs a high fish meal level in diet; even D1 is not good for that.
· Line 36, you add they experienced nutritional stress.
· When there is no significant difference, no need to report P value.
· Please change the treatment names to something meaningful. For example, Control, 25NGP, 50NGP, and 75NGP. Readers can understand your MS easier; Please make sure you updated the MS text, figures and tables with this change.
· The conclusion is not correct and even 25% still is not good enough to suggest.
· Line 51, is not correct!!! Please make sure you check the information before citing them.
· Line 75-119, please summarize it and only focus on fish meal studies with these proteins.
· Line 125 and elsewhere, please use the common name throughout the MS.
· Table 1, please add the basic protein to the table.
· Table 3, please highlight that in MS, the possible impaired growth is due to imbalanced fatty acids.
· Juvenile fish (approximately 2 g) used in this experiment were procured from a fish 175 havens!!!!!!!!!!! What!!!!!!!. Please make sure you revise the MS deeply. Many parts need English revision.
· Table 5, please use the complete name of the parameters instead of abbreviations. It will be easier for readers to understand. Use the complete name if you use less than 5 times a parameter.
· Please use more distinguishable colours for the plots.
· Line 357, none-grain proteins, does not make sense; it just simply says alternative proteins. Please update the MS with this change,
· Line 361 and other parts of the MS, please only focus on the alternative proteins that you used in this MS rather than any random protein resources.
· Here and elsewhere, report P uppercase and italic (P<0.05).
· Throughout the MS, if there is no significant difference, no need to report the P-value.
· Please reorder the keywords alphabetically and capitalize each word.
· Here and throughout the MS, please first mention the common name plus the scientific name, and for the rest of the MS, just report the common name.
· Please update the introduction with recent works in these alternative proteins as many studies are available from the last two years, which were not included in this section.
· Please review the literature much more carefully and cite more appropriate references.
· Please mention the novelty of your work in the last paragraph of the introduction.
· For each analysis, please clarify how many fish were taken.
· Some parts of the discussion are better updated with research in 2022 and 2023 as they refer to some old references. Please update the discussion with the latest studies as much as possible.
· Although you wrote this section well, you can still improve it by answering these questions and annotating them into the discussion section. Why were these results observed? Discuss more possible reasons.
· The conclusion needs to be revised and more comprehensive concepts should be added there.
Tables and Figures
• Please explain a little bit about your experimental treatments per each Table and Figure. Each Table and figure should represent enough information separately from the text.
• Double-check the units and titles of all Tables.
• Please mention in the footnote of all Tables which kind of statistical method you used for comparing the means.
When revising your manuscript, please consider all issues mentioned in the reviewers' comments carefully with clear outlines for every change made in response to their comments including suitable rebuttals for any comments you deem inappropriate. Please itemize your response to each review comment and highlight the revised at re-submission.
Kind regards
Moderate English revise
Author Response
请参阅附件。

Reviewer 2 Report
Comments and Suggestions for Authors
On the manuscript entitled “Dietary fishmeal can be partially replaced with non-grain compound proteins through evaluating the growth, biochemical indexes, and muscle quality in marine teleost Trachinotus ovatus”, with reference number animals-2388003, authors explored the inclusion of non-grain compound protein (NGCP) prepared from bovine bone meal, dephenolized cottonseed protein and blood cell meal an alternative source of protein for substituting FM in the diets of an aquaculture fish species, Golden pompano (Trachinotus ovatus).
With this purpose, authors tested 4 levels of NGCP inclusion (0, 25, 50 and 75%) in diets for juveniles during 65 days. The main finding/conclusion is that the present NGCP might be included at least 25% based on data obtained from tested parameters.
The issue here covered is interesting as try to solve one of the main bottlenecks on the sustainable growth and development of the aquaculture, reducing the dependency of fish meal (FM) in aquafeeds.
Although it is already known that non-grain proteins can be partially included in aquafeeds, the new source of non-grain proteins in compound form might benefit its implementation in the aquafeeds. Authors should address several issues before the present manuscript can be accepted for publication in Animals journal.
Introduction section:
Line 120-124: In these sentences, the authors have mentioned the objectives of their study for replacing FM in T. ovatus feed by NGCP (non-grain compound protein) by evaluating the growth index, proximate composition, serum biochemical indexes, muscle quality, and protein and lipid metabolism gene mRNA expression levels of fish in each treatment (D0-D3). Other lines (125-127) are related to the conclusion section, which is presented after performing the work and interpreting the results obtained from the research.
Materials and Methods section
Line 132: “an” NGCP should be corrected to “a” NGCP
Line 137: “NFCP” should be written as “NGCP”
Lines 141-142: “The diets specific ratios and basic nutrition has shown in Table 1” can be re-written as “The diets specific ratios and basic nutrition have been shown in Table 1”.
Lines 142-143: Based on the journal citation format, it should be mentioned the name of Ma et al. before [44]
Line 143: How long did it take to dry at 17 °C?
Line 143: Until “dry” is written as until “dried”
Line 149: The authors have already shown the ratio of 50, 30 and 20 for bovine bone meal, dephenolized cottonseed protein and blood cell in line 133, respectively. Which ratio is correct?
Moreover, NFCP is wrong and it should be replaced by NGCP.
Line 155: Which is the name of this VB: 2800 mg?
Line 162: This sentence “The dietary amino acid composition and fatty acid composition were showed in Table 2 and 3 “ should be re-written as “The dietary amino acid composition and fatty acid profile have been showed in Table 2 and 3, respectively.
Table 2: Amino acid Histidine (His) is listed under essential amino acid (EAA) profile while it is listed as non- essential amino acid (NEAA) in table 2. Therefore, the sum of the EAA and NEAA is changed in the table.
Line 176: In this section, it needs to be indicated the water quality of rearing condition like dissolved oxygen, salinity, pH and temperature by the authors.
Line 179: What was the name of Manufacturer Company and composition of commercial feed used by the juvenile?
Line 179: How long was the acclimatization period that the juvenile weight was increased from 2g to 9.71g!?
Lines 210-222 (subsection Amino acid and fatty acid composition assays): The samples including the feed and tissue have been used for determination of fatty acid composition. In this part, the authors should mention both samples. For example, in line 212, it points to “feed samples” and tissue sample is missed.
Comment: Why have the authors not detected the amino acid profile in the muscle?
Line 230: “to extracte” is wrong. It should be corrected as “to extract”.
Results section:
Lines 256-257: This sentence “furthermore, VSI and HSI of D1 group were significantly decreased compared to those of D2-D3 groups and that of D3 group, respectively (p ˂ 0.05)” has not clear statement. It should be re-written. VSI of D1 group was significantly decreased compared to that of D2-D3 groups while HSI of D1 showed a significant decrease compared to that of D3 group.
Line 272: In sentence of “and the whole-body ash and moisture content were significantly increased”, the moisture content is referred to muscle.
Line 275: Check space in this phrase “The above resultsindicate”
Table 6: Please make the whole body and muscle as Bold to see better.
Line 304: In sentence “……the D1 group were not were not significantly different (p > 0.05)……. the” the phrase “were not” is typed twice.
Line 306: The sentence “of D1 group showed significantly increased compared to the D2-D3 groups (p ˂ 0.05). The” Should be re-written. It shows an error in grammar ; what is the meaning of “ showed significantly increased”?
Line 338: In “ Figure 3. The mRNA expression levels of genes related to protein metabolism of juvenile T. ovatus” the scientific name should be had an italic format.
Line 347: In sentence” mRNA expression of fas, srebp1, and acc1 between all groups (p > 0.05). The above results” the “ between should be among.
Comment: As asked in section Materials and methods, why the amino acid profile was not determined in the muscle?
Discussion
Line 357: This sentence “Non-grain proteins that are widely available and inexpensive have the potential” can be re-written as “Non-grain proteins, being widely available and inexpensive, have the potential”
Lines 372-374: This research is conducted on juvenile T. ovatus and results obtained from this work may be used only about this species with stage of life “juvenile”. Therefore, authors can not expand their finding to other aquafeeds.
Lines 381-382: “…….of non-grain proteins mixed into fish feeds was superior compared to single proteins” is written as “…..of non-grain mixed proteins into fish feeds was superior compared to ones containing single proteins”.
Line 395: In”…. Serum TP—comprising ALB and GLB—has several physiological roles, including maintaining” such as is better to be used instead of “including”.
Line 416: In “…..their deficiency affects the growth of fish [64, 65]” affects should be corrected as affect.
Line 418-422: The authors have explained about the levels of dietary arginine and lysine and their effect on the expression of the target of rapamycin (TOR) pathway as a responsible role in nutritional status in fish. As previously mentioned in Results section, the authors have not determined the amino acid profile in juvenile muscle. Therefore, data from dietary amino acid may not alone interpret the above pathway.
Line 450: in “Similarly, as a study of grass carp (Ctenopharyngodon idella) shown” is written as “Similarly, as a study performed on grass carp (Ctenopharyngodon idella)”
Lines 463-467: As mentioned above, the amino acid profile from muscle can help to interpret these findings. The plant proteins are poor in essential amino acids like lysine and methionine that can be affected the muscle quality. Moreover, the most content of fatty acid in plants is n-6 PUFA that differs them from n-3 pufa and n-3 hufa found in fish meal.
Line 490: Data Availability Statement: …dates… is wrong. Data is correct.
Dear Editor in chief
The manuscript is written well and quality of English language is approved.
However, there are some errors in spelling and some errors in grammar that have been explained in Comments and Suggestions for Authors section.
Regards,
Round 2
Reviewer 1 Report
I suggest authors do a final check for errors and then be ready for further steps.
Good
Reviewer 2 Report
The manuscript has been revised according to requested comments and it can be accepted for publication.